# Drug2ways: Reasoning over causal paths in biological networks for drug discovery

**Daniel Rivas-Barragan**[1,2], **Sarah Mubeen**[3,4], **Francesc Guim Bernat**[5], **Martin Hofmann-Apitius**[3], **Daniel Domingo-Fernández**[3,4]*

**1** Barcelona Supercomputing Center, Barcelona, Spain, **2** Computer Architecture Department, Universitat Politècnica de Catalunya, Barcelona, Spain, **3** Department of Bioinformatics, Fraunhofer Institute for Algorithms and Scientific Computing, Sankt Augustin, Germany, **4** Fraunhofer Center for Machine Learning, Germany, **5** Intel Corporation Iberia, Madrid, Spain

* daniel.domingo.fernandez@scai.fraunhofer.de

**Data Availability Statement:** The data (i.e., networks) used in the case scenarios of the manuscript can be found at https://github.com/

## Abstract

Elucidating the causal mechanisms responsible for disease can reveal potential therapeutic targets for pharmacological intervention and, accordingly, guide drug repositioning and discovery. In essence, the topology of a network can reveal the impact a drug candidate may have on a given biological state, leading the way for enhanced disease characterization and the design of advanced therapies. Network-based approaches, in particular, are highly suited for these purposes as they hold the capacity to identify the molecular mechanisms underlying disease. Here, we present drug2ways, a novel methodology that leverages multi-modal causal networks for predicting drug candidates. Drug2ways implements an efficient algorithm which reasons over causal paths in large-scale biological networks to propose drug candidates for a given disease. We validate our approach using clinical trial information and demonstrate how drug2ways can be used for multiple applications to identify: i) single-target drug candidates, ii) candidates with polypharmacological properties that can optimize multiple targets, and iii) candidates for combination therapy. Finally, we make drug2ways available to the scientific community as a Python package that enables conducting these applications on multiple standard network formats.

## Author summary

At any given time, a large set of biomolecules are interacting in ways that give rise to the normal functioning of a cell. By representing biological interactions as networks, we can reconstruct the complex molecular mechanisms that govern the physiology of a cell. These networks can then be analyzed to understand where the system fails and how that can give rise to disease. Similarly, using computational methods, we can also enrich these networks with drugs, diseases and disease phenotypes to estimate how a drug, or a combination of drugs, would behave in a system and whether it can be used to treat or alleviate the symptoms of a disease. In this paper, we present drug2ways, a novel methodology designed for drug discovery applications, that exploits the information contained in a biological network comprising causal relations between drugs, proteins, and diseases.

drug2ways/drug2ways along with the code, all of which are openly available.

**Funding:** This work was developed in the Fraunhofer Cluster of Excellence "Cognitive Internet Technologies"(SM and DDF). The funders had no role in study design, data collection and analysis, decision to publish, or preparation of the manuscript.

**Competing interests:** The authors have declared that no competing interests exist.

Employing these networks and an efficient algorithm, drugways2 traverses over the ensemble of paths between a drug and a disease to propose the drugs that are most likely to cure the disease based on the information contained in the network. We hypothesize that this ensemble of paths could be used to simulate the mechanism of action of a drug and the directionality inferred through these paths could be used as a proxy to identify drug candidates. Through several experiments, we demonstrate how drug2ways can be used to find novel ways of using existing drugs, identify drug candidates, optimize treatments by targeting multiple disease phenotypes, and propose combination therapies. Owing to the generalizability of the algorithm and the accompanying software, we ambition that drug2ways could be applied to a variety of biological networks to generate new hypotheses for drug discovery and a better understanding of their mechanisms of action.

This is a *PLOS Computational Biology* Methods paper.

## Introduction

Biological processes principally arise from interactions linking discrete biological entities. Far more rare, however, are processes that can be attributed to entities functioning in isolation. Hence, elucidating sets of interactions between biological entities is essential in understanding the mechanisms governing health and disease. Given the vast number of interactions that can occur in a particular biological system, these interactions are often abstracted and organized into large and highly interconnected computational networks. Many of the basic principles and methods from graph theory tend to be well-suited for network biology and applicable to various network types, such as protein-protein interaction (PPI), gene regulatory, and signalling networks [1]). Several discrete models, such as logical models [2] and Boolean networks [3,4] are common choices for their qualitative representation.

In a generic biological network representation, nodes denote entities, while edges denote their interactions. Multimodal networks can capture a wide range of biological scales, including physical entities (e.g., genes, proteins, and metabolites) or higher order concepts (e.g., biological processes, phenotypes, and diseases). Causal edges are those that possess directionality through direct interactions or through intermediaries [5]. These connections frequently occur in gene regulatory and metabolic/biochemical networks, while undirected edges are commonly present in chemical similarity or PPI networks to, for instance, represent symmetric binding relationships. For the latter group of edges, several methods [6,7] have emerged to assign directionality to interaction pairs (e.g., characterizing regulatory relationships as activation or inhibition relations) in order to assert causality which can be useful for various purposes. An example lies in discerning whether causal interactions between a drug target and intermediary proteins will inhibit a certain phenotype, a drug's intended effect, or activate it instead. Taken together, these networks enable a wide range of applications such as identifying disease mechanisms [8], making predictions on network perturbations [9], facilitating pathway analyses [10], establishing novel therapeutic drugs [11], and drug repurposing to detect potential therapeutic candidates [12].

Drug discovery is a major application that particularly benefits from network-based methods [11]. Typically, the traditional approach to drug discovery is characterized as follows: a drug target is selected based on an expressed phenotype, an assay is prepared for the target, high throughput screening (HTS) is performed, and hit or lead compounds are identified [13].

Though it may be the more conventional approach, the process tends to be laborious and is associated with both high costs and attrition rates. The latter can be attributed to several factors; firstly, experiments demonstrating the efficacy of drugs through their specific binding to a target may not be reproducible *in vivo* given the compartmentalization of the cell and/or the potential for other binding partners [14]. Secondly, in failing to investigate the cause of dysfunction that leads to disease within an appropriate biological context (e.g., molecular, cellular, or disease), the design of drugs is arbitrary [15]. These issues represent some of the prototypic problems that network-based approaches are ideally suited to address.

Beyond the utility of network-based methods for single target drug discovery and repurposing, these methods are also increasingly being used for the identification of pharmacological interventions that reverse multiple pathological states and in the design of drug combinations [16]. Although certain aspects of a pathology may be corrected by a single target drug, a multi-target drug or drug combination approach can have greater efficacy in reversing a disease or an expressed phenotype [17]. By taking into account causal mechanisms, network-based approaches can identify multiple targets within a network which, when modulated, can elicit synergistic effects[18]. Notably, combination therapies have successfully been used for several disease conditions including cancers [19,20] and the symptomatic management of Alzheimer's disease [21].

Various attributes of biological networks can serve as viable measures for network-based drug discovery. For instance, proximity measures such as the shortest path between a drug profile and a disease module have been used to identify potential drug repurposing candidates [22,23]. Additionally, centrality measures such as closeness and betweenness centrality also consider the shortest paths between pairs of nodes in order to pinpoint initial drug candidates [24,25]. However, potentially therapeutic targets may be connected to disease-relevant genes through paths not accounted for when solely considering shortest paths. Nevertheless, approaches which use non-shortest paths along a network are not without their limitations; as the size and complexity of networks increase, so too do the number of possible paths that can be traversed through the network, requiring greater computational power. Similarly, with an increasing number of nodes and edges, identifying multiple drugs for combination therapies that simultaneously target multiple disease-relevant genes and/or mitigate side-effects, can suffer from combinatorial explosions. Furthermore, not all paths in a network may be biologically plausible; erroneous interactions and those which are not biologically-relevant may also be present. Thus, making predictions for single and combination drug therapies can become highly challenging.

Here, we present drug2ways, a novel methodology applied to multimodal causal networks for the prediction of new drugs and the repurposing of existing ones. Our methodology consists of two main steps which jointly aim to address the high computational demands required to traverse large-scale, biological networks and to apply a reasoned approach to propose drug candidates for new indications by inferring causal paths. Firstly, drug2ways leverages a sophisticated and efficient algorithm to calculate all paths up to a given length between a drug and a disease or a set of phenotypes. Secondly, drug2ways traverses these paths to propose the set of drugs that are most likely to generate a desired phenotypic change. We demonstrate the utility of drug2ways for three different applications in order to identify: i) potential drug candidates, ii) potential candidates that optimize multiple target nodes of interest (i.e., indications and phenotypes) and iii) candidates for combination therapy. Finally, we make drug2ways available to the bioinformatics community as a Python package (https://github.com/drug2ways) that enables conducting the aforementioned applications on multiple standard network formats.

## Results

We ambition multiple applications for drug2ways (Fig 1) which we present in three case scenarios and validate in two independent networks, the OpenBioLink knowledge graph (KG) and an In-House network. In the Subsection *Identifying drug candidates*, we first validate our methodology by showing how it can be used to identify potential drug candidates for various indications, while in the Subsection *Identifying drug candidates with multiple phenotypic targets*, we demonstrate how drug2ways can identify drugs that target sets of phenotypes present in specific indications. Finally, in the Subsection *Proposing combination therapies*, we show its utility in finding potentially efficacious drug combinations for combination therapy. In each of the three applications, the problem can be generalized to finding the relative effect of all paths between nodes representing chemicals and nodes representing phenotypes or clinical manifestations. Each application consists of reasoning over all possible paths of a predetermined length to evaluate the efficacy of either one or more chemicals in reverting the target node of interest (i.e., a manifestation and/or a set of associated phenotypes). This task can be conceived of as a brute-force search for all drugs and indications/phenotypes in a network for a given range of path lengths in order to prioritize drug candidates for each of the target nodes of interest.

The drug2ways algorithm incorporates two variants, namely all paths (i.e., a path in which repetition of vertices occurs) and simple paths (i.e., a path in which all vertices are distinct (and therefore, all edges)), enabling users to account for or ignore feedback loops (i.e., cycles), respectively (Fig 1D). Each of these three applications is associated with a high computational cost, especially the latter two which require calculations of a higher degree of complexity to identify potential candidates with multiple phenotypic/disease targets. However, because of the efficient implementation of the algorithm, each of these applications is attainable, which we demonstrate in the Subsection *Performance comparison and scalability of the algorithm* where we finally explore the scalability of drug2ways and compare it with standard path-finding implementations.

### Identifying drug candidates

In Table 1, we summarize the results of drug2ways in recovering clinically-investigated drug-disease pairs for the top-ranked candidates in each of the validation experiments. Firstly, for both the original networks, drug2ways was able to retrieve a large proportion of drug-disease pairs that have been tested in clinical trials by calculating all paths up to a given length between a drug and an indication (i.e., $lmax$), although both networks exhibited differences based on the prioritization criteria described in the Subsection *Validation experiments*. For instance, the most restrictive prioritization criteria (i.e., 7/7 $lmax$ inhibited the disease) yielded the best results for the In-House network, recovering nearly 40% of true positives from all prioritized pairs in the top-ranked list for all paths and simple paths respectively, while OpenBioLink yielded no prioritized pairs altogether. However, after a minimum relaxation of the prioritization criteria (i.e., 6/7 $lmax$ inhibited the disease), OpenBioLink showed good results (i.e., ~50% and ~10% recovery rate for all paths and simple paths, respectively) while the recovery rate decreased for the In-House network to approximately 12%. In comparison, the proportion of true positives with respect to all possible combinations of drug-disease pairs from ClinicalTrials.gov is 3.19% for OpenBioLink (5.151/161.040) and 3.76% (9.537/253.638) for the In-House network, highlighting the significance of our results. These proportions are equivalent to the probability of randomly picking a true positive, which is comparatively much lower than the results yielded by drug2ways. In contrast, drug2ways failed to recover any true positives from the permuted versions of the original networks, further highlighting the validity of the results from the original networks.

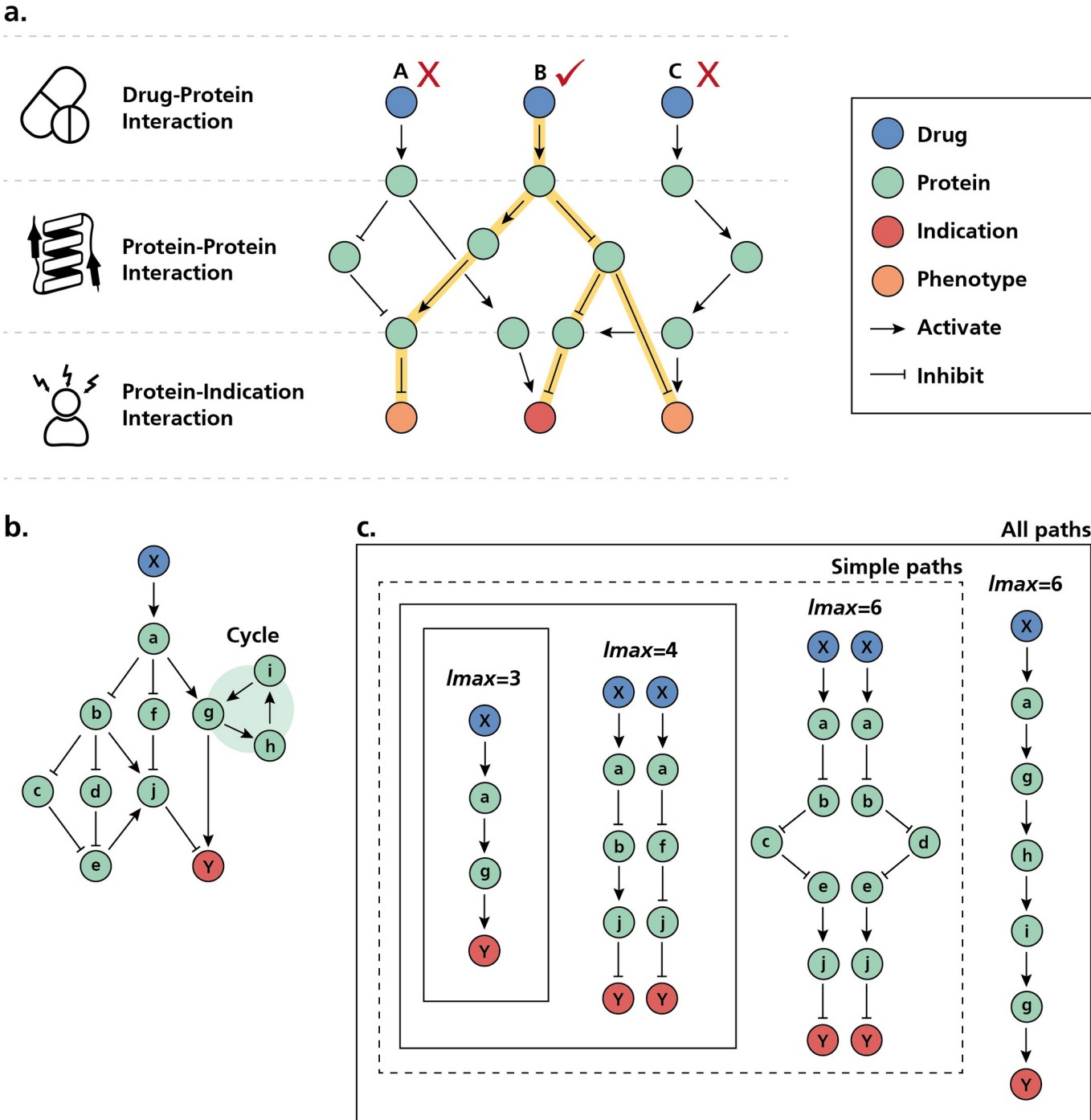

**Fig 1. Schematic illustration of causal reasoning by drug2ways over simplified networks. a)** Prototypic network used by drug2ways for drug discovery. The network contains causal relations between three modalities (i.e., drugs, proteins, and indications/phenotypes). Here, singular paths from three drugs to an indication as well as associated phenotypes are shown, though a single drug may contain multiple paths to a given indication/phenotype. Drug2ways reasons over all possible paths in a network between a drug and an indication/phenotype to predict the relative effect of each drug. In the example, we want to investigate whether one of the three drugs depicted inhibits an indication and its two phenotypes. While all three drugs target the disease, two of the three (i.e., drug A and C) fail to produce the desired effects (i.e., inhibition of the indication of interest and its two associated phenotypes). By reasoning over all the paths between the drug and the three target nodes of interest (i.e., indication and its phenotypes), drug2ways predicts that drug B could be a promising candidate as the majority of the paths would result in their inhibition, and thus produce a therapeutic effect. Similarly, drug2ways can also be used to evaluate the effect of a drug on a single indication/phenotype or to assess the effect of drug combinations. **b)** Example network containing all paths between a given drug and an indication. **c)** All possible paths between the drug and indication in **(b)**. The drug2ways algorithm incorporates two variants, namely *all paths* and *simple paths*, enabling users to account for or ignore feedback loops (i.e., cycles), respectively. We distinguish between different paths based on the maximum number of allowable edges from a

drug X to an indication Y (i.e., *lmax* parameter). For instance, the shortest path between the drug and the indication has an *lmax* of 3 while an *lmax* of 6 will capture this and four additional simple paths, two of length 4 and a further two of length 6. Using the *all paths* version of the algorithm, an additional cyclic path of length 6 is also captured.

Given the small-world property of most biological networks, the predominant approaches in network-based drug discovery tend to investigate the shortest path between a drug and a disease. We thus compared our method to the shortest-path approach. While the results obtained using the shortest path are better than random, the shortest path tends to return a relatively high number of candidate pairs (>5.000) and a significantly lower recovery rate (~8%) than drug2ways (Table 1). Furthermore, we studied the lengths of the paths of candidate pairs prioritized by the shortest-path approach and, as expected, found that the vast majority of the paths are of lengths less than 4 (S1 Fig). In fact, the majority of the paths are *lmax = 2*, which corresponds to a direct drug-target-disease path. This indicates that the shortest-path approach can overlook diseases that are distant from drug targets, potentially explaining the difference in recovery rate between shortest-paths and drug2ways. Furthermore, while the shortest path only accounts for a single path between a drug and a disease, as an additional experiment, we investigated the total number of paths between all drug-disease pairs calculated from drug2-ways using *lmax = 8* to verify that predictions were not driven by the existence of a single path but by the directionality inferred through the ensemble of all paths (S2 Fig). We found that a large number of paths were present between most of the drug disease pairs, which when taken into account, could also explain the difference in the recovery rate.

The criteria selected for validation focused on prioritizing pairs exhibiting consistent scores (i.e., activation/inhibition ratio) through a wide range of *lmax*. In selecting this criteria, we intended to prevent any influence of path length (i.e., *lmax*) on the results. As expected, the results also indicate that the *lmax* parameter and the prioritization criteria should be adapted for each new network. Thus, we recommend that users that intend to apply our methodology on their own networks follow a similar approach by using a broad range of *lmax*. Beyond the configuration of the *lmax* parameter, we also recommend tuning a threshold value representing the relative effect of the drug on the indication, gradually decreasing this value to include additional, potential drug candidates. In this way, the Python implementation of drug2ways enables users to configure their experiments contingent upon the particular characteristics of the network (e.g., content and size).

Due to a lack of information on the directionality of protein-disease relations from high-quality resources, while generating both networks, we inferred association edges from DisGe-Net [26] as activation edges (see Methods). Such a strong assumption implies that all proteins have an activation effect on the disease and ignores the possible inhibitory effects some of these proteins may have. Accordingly, due to this arbitrary inference, we hypothesized that

**Table 1. Results of the validation experiments.** The table presents the validation experiments for each of the four networks (i..e, OpenBioLink, permuted OpenBioLink, In-House, and permuted In-House) using two variants of the algorithm (i.e., all paths and simple paths) based on two different prioritization criteria (**see Methods**) as well as the results yielded when only considering the shortest path between a drug-disease pair. For each experiment, we report the relative number of true positives in the list of drug-disease pairs prioritized by drug2ways. The proportion of true positives recovered by both variants of drug2ways in the two original networks are significantly higher than chance level (i.e., 3.19% for OpenBioLink and 3.76% for the In-House network).

| Network | All Paths | | Simple Paths | | Shortest Path |
|---|---|---|---|---|---|
| - | **7/7 Inhibit** | **6/7 Inhibit** | **7/7 Inhibit** | **6/7 Inhibit** | **-** |
| OpenBioLink | 0/0 (%) | 2/4 **(50%)** | 0/0 (%) | 1/11 **(9.09%)** | 381/5.130 **(7.43%)** |
| Permuted OpenBioLink | 0/0 (%) | 0/0 (0%) | 0/0 (0%) | 0/0 (0%) | 40/5.130 **(0.78%)** |
| In-House | 20/53 **(37.74%)** | 105/919 **(11.43%)** | 22/54 **(40.74%)** | 106/872 **(12.16%)** | 807/9.537 **(8.46%)** |
| Permuted In-House | 0/0 (0%) | 0/6 (0%) | 0/0 (0%) | 0/7 (0%) | 274/9.537 **(2.87%)** |

some of the drug-disease pairs predicted as activating may indeed represent the opposite sign and also represent potential drug candidates. Thus, besides investigating drug-disease pairs that were consistently inhibited, we were also prompted to investigate pairs that were consistently activated. Confirming our hypothesis, we found that although based on our criteria, relatively few pairs were prioritized, clinically-investigated drug-disease pairs were also highly represented among the top-ranked active pairs (S3 Table).

In summary, our findings demonstrate the ability of drug2ways to recover a high proportion of clinically-tested drug-disease pairs. Due to our network design, candidate pairs consistently aggregate at both extremes of the distribution regardless of the relative directionality given by the ensemble of paths. Finally, among the novel drug-disease pairs that have not yet been tested in clinical trials, we have found multiple combinations reported in the literature, thus alluding to the potential for many other promising candidates for drug discovery that could be worth further exploring.

## Identifying drug candidates with multiple phenotypic targets

The identification of drugs with several target nodes of interest (i.e., indications/phenotypes) can lead to more efficacious treatments, albeit their discovery is far more complex and thus represents a greater challenge than single-target drugs. In practice, this application is highly relevant as disease conditions can often manifest as sets of phenotypes. While the previous subsection demonstrated how our methodology is capable of identifying interesting single target drug candidates, in this subsection, we demonstrate how a network-based method can identify drug candidates that optimize multiple disease and phenotypic targets.

Here, we manually selected an indication and associated phenotypes present in both the In-House and OpenBioLink networks (S4 Table). Fig 2A illustrates the results of running the all

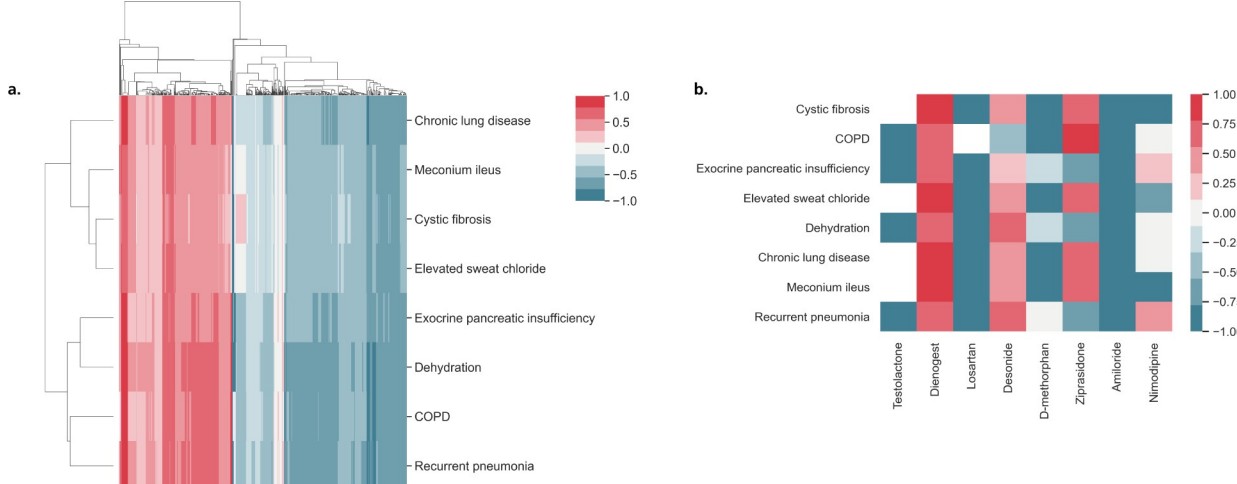

**Fig 2. Identification of drugs targeting an indication and several associated phenotypes.** The heatmaps summarize the results of running the all paths version of the drug2ways algorithm over the In-House network for variable path lengths. While the algorithm outputs scores between 0 and 1, where 0 denotes no activation or inhibition and 1 denotes a full activation or inhibition, scores were normalized between the range of -1 to 1. Here, normalized scores of the relative effects of drugs on cystic fibrosis and several of its associated phenotypes are displayed where values below and above 0 denote the inhibition (blue) and activation (red) of all paths between a drug and target indication/phenotype at a specific *lmax*, respectively, whilst 0 denotes a cancelling effect (gray). In a fourth case, no paths exist between the drug and indication/phenotype (white). **a)** Hierarchical clustering of normalized scores of the relative effects of all drugs in the In-House network on cystic fibrosis and related phenotypes at *lmax 8*. **b)** Heatmap illustrating a subset of drugs at *lmax 4* which distinctly optimize therapeutic effects through inhibition of several disease/phenotypic targets (e.g., Amiloride, D-methorphan, Losartan), activate the disease and/or its phenotypes (e.g., Dienogest), result in both the inhibition of some diseases/phenotypes and the activation of others (e.g., Desonide, Ziprasidone, Nimodipine), or do not possess paths to particular targets (e.g., Testolactone).

paths version of the drug2ways algorithm over the In-House network at an *lmax* of 8 for cystic fibrosis (CF) and seven related phenotypes. The heatmap shows that in selecting larger values of *lmax*, the vast majority of drugs (i.e., 626/671 drugs in the In-House network also in ClinicalTrials.gov) possess paths to each of the targets. We also note that most drugs in the network affect the indication and the phenotypes in a given direction (e.g., inhibition), while only a small minority will result in the activation of some phenotypes and/or indication and in the inhibition of others.

Once again, we altered the value of *lmax* between 2 and 8 to investigate the relative effects of drugs yielded with varying path lengths. While beyond *lmax 4*, we found little variation in the number of drugs containing paths to at least one target indication/phenotype (ranging from 602 drugs at *lmax 5* to 626 drugs at *lmax 8*), we found fewer drugs at and below *lmax 4* (i.e., 55 at *lmax 2*, 234 at *lmax 3*, and 539 at *lmax 4*). Fig 2B illustrates a subset of drugs at *lmax 4* that reverse, increase, cause no effect or have no paths to the indication and/or phenotypes. Among these drugs, we further investigated losartan, a drug under investigation in clinical trials for CF and studied the proteins implicated in paths of maximum length 4 between this drug and the disease. These proteins included *AGTR1*, whose reduced activity by pharmacological intervention has resulted in improved pulmonary functioning in mice with CF [27], and *TGFB1*, reduction of which by losartan has been shown to reverse mucociliary dysfunction related to inflammation and CF in animal models [28].

## Proposing combination therapies

Combination therapies have been gaining major consideration for the treatment of disease and management of symptoms through the modulation of several targets by multiple drugs. However, with each additional drug for combination therapy, the task of identifying efficacious combinations by a network-based approach can result in a substantial increase in computational complexity, thus requiring efficient algorithms. Therefore, we were prompted to utilize drug2ways in a further application to explore the predicted effects of a combination of drugs on a given indication. We identified drug combinations consisting of pairs of drugs, though would like to note that our method could be used to identify combinations involving any number of drugs.

We manually selected several cancer types (i.e., breast cancer, colorectal cancer, lung cancer and melanoma) present in our In-House network to demonstrate an additional application of drug2ways to predict potential drugs for combination therapy. Similar to the previous two applications, as an input, we only considered drugs in the In-House network that were also present in ClinicalTrials.gov and used drug2ways to propose drug combinations at *lmax* 4. For each of the four cancer subtypes, we then investigated existing drugs for their management and identified those that were also present in our network. We then focused on drug combinations that contained these drugs and caused inhibition of the cancer subtype. Table 2 lists a

**Table 2. Examples of predicted combination therapies supported by literature evidence on four cancer types.** The table reports drug combinations identified by drug2ways that inhibit each of the various cancer types and supporting literature evidence. These results were obtained by running the all paths version of the algorithm over the In-House network for *lmax 4*.

| Cancer type | Drug 1 | Drug 2 | Evidence |
|---|---|---|---|
| Breast cancer | Palbociclib | HCQ | [29] |
| Breast cancer | Palbociclib | Tamoxifen | [30] |
| Colorectal cancer | Palbociclib | Trametinib | [31] |
| Lung cancer | Dabrafenib | Trametinib | [32] |
| Lung cancer | Palbociclib | Trametinib | [33] |
| Melanoma | Mebendazole | Trametinib | [34] |

subset of drug combinations proposed by our methodology to inhibit specific cancer types and literature evidence on their potential therapeutic effects.

While, here we have only discussed drug combinations already in clinical trials or with correspondence to the literature, a multitude of combinations identified by our methodology that could potentially inhibit a disease but have not been reported thus far, represent potentially efficacious, novel combination therapies. Additionally, while in showcasing this functionality of our method, we have used all possible combinations of drugs that are both in our network and in clinical trials, this application can also be performed with a smaller set of drugs to evaluate the effect of particular drug combinations on a given set of diseases and/or phenotypes. Finally, each of the paths between a drug-disease pair can be defined as a sub-network representing biological processes and using pathway enrichment methods implemented in drug2-ways, the mechanism of action of the drug can be elucidated.

## Performance comparison and scalability of the algorithm

The applications described above have been conducted on large-scale networks comprising tens of thousands of nodes and edges, yet the size of biological networks can increase to incorporate millions. Therefore, the implementation of the algorithm has been designed to maximize its performance. Here, we compared drug2ways to the Python NetworkX library [35] (https://networkx.github.io/" https://networkx.github.io/) and the C++/Python NetworKit library [36] (https://networkit.github.io/). We compare drug2ways against these two libraries, as both are widely used and already implement optimized methods for graph traversal and path retrieval. Both libraries implement a method to obtain all simple paths in a graph with a maximum path length. Fig 3 illustrates the runtime of each network-method pair in logarithmic scale on the y-axis, (i.e. for each network-method pair, the figure shows the time to count activation and inhibitory paths for each drug-disease pair in the network). As expected, the runtime is heavily dependent on the maximum path length *lmax* that we want to analyze. We added a timecap of 1.000 seconds (i.e. around 16 minutes) to the experiments, which is enough to show the method's scalability trendline and its exponential growth, while beyond this

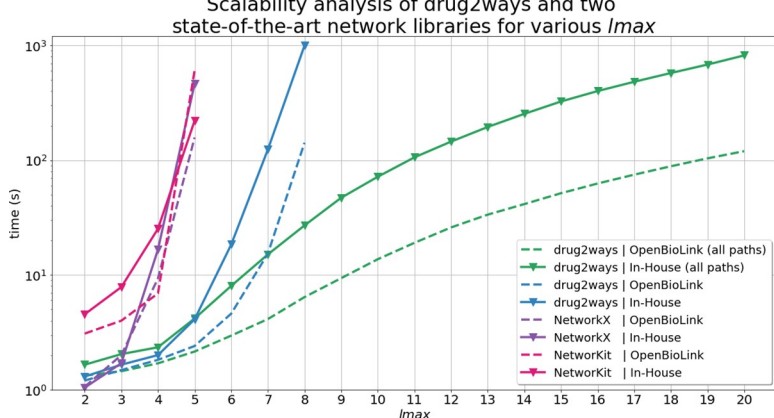

**Fig 3. Average time required to calculate the effect of simple paths for all drug-disease pairs used in the validation on two heterogeneous networks using different *lmax*.** The analysis was also conducted to take paths with repetitions of vertices between drug-disease pairs into account using the *all_paths* variant of drug2ways, but not for the NetworkX and NetworKit libraries which lack equivalent implementations. Nevertheless, the implementations of both libraries could be easily adapted to return paths with repetitions of vertices. However, without the proper optimizations described in the Subsection *Theoretical background*, these would have a higher complexity than their *all_simple_paths* counterpart as nodes would be revisited. Therefore, for both libraries we use simple paths as the baseline for the analysis.

timeframe, the runtime becomes unreasonably high. All three methods to count simple paths show a clear exponential growth in runtime. However, while NetworkX and NetworKit can be run with up to an *lmax* of 5, drug2ways with simple paths is several orders of magnitude faster and is able to be run with up to an *lmax* of 8. The comparison also shows that of the three different methodologies, only drug2ways can be scaled for large values of *lmax* on both versions of the algorithm.

The *all_paths* variant of drug2ways does not show a pronounced exponential increase in time. However, the *all_simple_paths* variant shows a pronounced exponential increase in running time as it is computationally more expensive than *all_paths*. Here, the two standard libraries show a rapid exponential increase in time with *lmax* values as low as 4 while drug2ways does not show a marked increase until values of *lmax* beyond 7.

Taken together, we can see how the *all_paths* variant can be easily used for any large-scale network even when values of *lmax* exceed 20, while the *all_simple_paths* variant requires both extensive computational power and time when such high values of *lmax* are reached. In contrast, it is impractical to run experiments on large *lmax* values using the other two standard libraries as they have not been optimized for the specific reasoning tasks presented in this work. Thus, these standard libraries would suffer from a high computation cost in conducting the applications of this approach (i.e., optimization of several phenotypes and/or an indication and identification of candidates for combination therapy), and in calculating paths on high values of *lmax*. Finally, we would like to note that in order to conduct a fair comparison, the experiments presented have not been conducted using the parallelization feature of drug2ways. Thus, we expect that in using this feature for the analysis, the difference in the performance between drug2ways and the other two libraries would have been even more pronounced.

## Discussion

Increasingly, network-based methods are emerging as promising alternatives to traditional approaches for drug discovery by taking into account causal mechanisms responsible for disease. Here, we have presented a robust and efficient method that leverages causal interactions in biological networks to predict drug candidates for a given disease or a set of phenotypes, as well as pairs of drugs for combination therapy. While previous methods have focused on leveraging network proximity methods (e.g., shortest paths) between drugs and indications [22,23], drug2ways leverages all the paths between a given drug and disease. Although not all paths in a network may be plausible as some paths may be irrelevant or erroneous, we hypothesize that by reasoning over a multitude of possible paths, we can estimate the relative effect of each drug on a disease as the average of all possible paths. In doing so, we assume that a drug has a greater likelihood of modulating a disease as the number of possible paths connecting a drug to a disease increases. Therefore, exploring all paths in which a drug could modulate a disease or a phenotype can serve as a proxy for the prediction of novel drugs. To test our hypothesis, we systematically predicted the effect of each drug on all diseases in two multimodal networks of different size and content. Next, we validated our results against clinical trial information showing that our approach could retrieve a large proportion of true positives. Furthermore, with a second application, we demonstrate the ability of our approach to identify single drugs that can simultaneously modulate multiple targets to revert a set of phenotypes. Finally, the third application shows how a similar strategy can be applied for combination therapy.

Although drug2ways requires multimodal networks that contain causal relations between drugs, proteins, and indications/phenotypes, it can also be tuned and applied to other networks with different properties. For instance, we propose the use of networks comprising non-molecular nodes, such as biological processes, in cases when molecular information is not

widely available. Given the exponential increase in computational complexity when using the algorithm on multiple drugs for combination therapy, we demonstrate this application exclusively on drug pairs. Nonetheless, the high performance of drug2ways, which also allows for parallelization, enables users to conduct experiments upon millions of combinations in contrast to other state-of-the-art network libraries which would require an immense amount of time and computational resources.

One of the major limitations of this work is the absence of signed causal information regarding the effect of proteins on indications and phenotypes. To circumvent this issue, we inferred all protein-indication and protein-phenotype associations as activations, an assumption that may not correspond to the true biology. Thus, due to a lack of such information, curating and qualifying directionality for these relations could be a future improvement for drug2ways. Additionally, we would like to acknowledge the possible effects of feedforward loops on the results, especially as *lmax* increases. However, the design of our validation has taken this factor into consideration. Finally, although we validated our results with clinical trial information and tested the robustness of our approach, by simplifying biology to a network of binary causal relationships, we overlook its quantitative aspects. Therefore, we would like to note that quantitative measures, such as kinetic rates for reactions, the confidence of the interaction, and the magnitude of the effect, may provide a more realistic representation and thus, could be considered in future work by adding these aspects as weights to the edges in a network. Finally, we also intend to investigate the feasibility of drug2ways to identify drugs that mimic disease phenotypes and hence, could be potentially employed to generate *in vitro* or *in vivo* models.

In summary, our approach demonstrates that reasoning over multiple causal paths in biological networks can potentially serve to predict candidates for drug discovery. From a translational perspective, drug2ways can be used to identify novel drugs and combination therapies for indications where their mechanisms of action can be well represented in a network. Finally, we provide a user-friendly Python package that enables conducting the three presented applications on biological networks in multiple standard formats.

## Methods

In the first four subsections, we outline relevant graph theoretical concepts, describe the graph traversal algorithm presented in the study, delineate its complexity, and provide details on the implementation of the software. Next, we discuss applications of the algorithm which are illustrated in case scenarios and validation experiments. In the final subsection, we provide details on the hardware used.

### Theoretical background

Given that most biological networks display the small-world property in which paths between pairs of nodes are relatively short, many genes can be in the vicinity of disease-relevant ones [14]. Accordingly, a simple yet effective approach to identifying potential drug targets is to consider nodes that are in close proximity to disease genes. However, not all of these nodes may necessarily be linked to disease genes, but rather, may simply be false positives resulting from spurious or irrelevant interactions [37]. Furthermore, such an approach can overlook interesting genes linked to disease-relevant ones by longer, alternative paths. One possible solution to this problem lies in traversing all possible paths between a pair of nodes to reach beyond the limits of local, proximity-based approaches. Beyond calculation of all paths between a drug and disease-related gene, however, a reasoned approach can be used to suggest how a drug may modulate a disease given the number of paths and types of interactions

between the two. Essentially, with a causal network containing directed relationships, signed -1 to indicate inhibition and +1 to indicate activation, we can define the relative effect of each drug as the proportion of activatory/inhibitory paths from all possible paths between the two (Fig 1). Nonetheless, with several thousand drugs and diseases, the computational complexity to traverse all possible paths between each pairwise combination can increase dramatically.

An intuitive solution to determine the relative effect of a drug on an indication would be to first find the set of all paths between them and then compute the effect on each of these paths. However, the problem of finding all paths in a network, which we will interchangeably refer to as a graph, is known to be NP-Hard (i.e., computationally hard), which are the class of problems in computational complexity that are not solvable in polynomial time. This makes the problem intractable as with an increasing number of vertices for some types of graphs (e.g., fully connected graphs), the total number of paths grows exponentially. However, to solve this problem we are not required to store the whole sequence of edges forming each path. Instead, if edges in a path are represented by their effects (i.e., -1 and +1 labels indicating inhibition and activation, respectively), we can define the combined effect of the path as the product of all edges it contains, while for the same set of edges regardless of the order they appear in the graph, the combined effect will always remain the same. This enables a series of optimizations which allow us to reduce time and space complexity, as explained in detail in the Subsection *Algorithm*. If a graph contains cycles (i.e., feedback loops), an infinite number of possible paths can be found by repeating the sequence of edges containing the cycle (Fig 1B). However, an increasing number of possible edges can also lead to an exponential increase in the number of paths, most of which may not be biologically plausible and result in the true biological effect becoming lost. We thus consider paths only up to a maximum length to limit the influence of cycles and highly elongated paths whilst still capturing feedback loops (Fig 1C).

We first define a series of terms that will be used throughout this section to provide a formal definition of the problem. Given an unweighted directed graph $G = (V, E)$, $V$ is the set of vertices (interchangeably nodes) and $E$ is the set of edges in the graph. A path is defined as a sequence of edges $(e_1, e_2, \ldots, e_k)$ that joins a sequence of vertices $(v_1, v_2, \ldots, v_{k+1})$ in a graph, for $1 \leq k \leq |E|$ such that $e_i = \{v_i, v_{i+1}\}$, for $1 \leq i \leq k$, where $k$ is the number of edges and the length of the path. Consequently, we denote a path between a source node $s$ and a target node $t$ as $p_{s,t}$, for $s$, $t \in V$ i.e. for the set of nodes $(v_1, v_2, \ldots, v_{k+1})$ joined by the path, $s = v_1$ and $t = v_{k+1}$, while nodes $v_i$, for $1 \leq i \leq k+1$ are *intermediate nodes* (see Table 3 for key definitions). Similarly, a *cyclic path* is a path when the first and last vertices it joins are the same, while a *simple path* is a path where all vertices are distinct. Furthermore, any edge $e \in E$ in $G$ represents a relationship between the pair of nodes it connects and it is labeled +1 or -1 depending on whether it is an activatory or an inhibitory relationship, respectively. Following, the effect that a node $s \in V$ has on node $t \in V$ over a given path $p_{s,t}$ is computed as $effect(p_{s,t}) = \prod_{i=1}^{k} e_i, \forall e \in p_{s,t}$, where $e_i \in \{-1, +1\}$ and is the label

**Table 3. Definitions of terms used in this paper.**

| Term | Definition |
|---|---|
| Simple path | A path in which all vertices are distinct (and therefore, all edges). |
| Cyclic path | A path in which repetition of vertices occurs. |
| All paths | The set of all paths, including those which contain cycles. |
| Intermediate node | Any node $v$ in a path between two nodes $u$, $t$, s.t. $v \notin \{u, t\}$. |
| Path length | The number of edges in a path between a source node and a target node. |
| *lmax* | The maximum length of the paths between a source and target node. In other words, for any given *lmax*, only paths with a length less than or equal to *lmax* are considered. |

of the $i^{th}$ edge in the path. A path $p_{s,t}$ is said to be an activatory path if its effect is equal to +1. Analogously, the path is said to be an inhibitory path if its effect is equal to -1.

Before defining the problem, we would like to remark that $p_{s,t}$ does not necessarily represent a singular path; as $s$ and $t$ might be connected by multiple sets of edges and different sets of edges may yield different effects between the nodes, a path is uniquely identified if its entire sequence of edges is unique. Furthermore, we would also like to remark that once the effect of a path is computed, we are no longer interested in the set of edges and intermediate nodes of a given path. Therefore, for simplicity, we define $P_{s,t}$ as the set of all paths between $s$ and $t$. Similarly, $A_{s,t}$ denotes the set of all activatory paths between $s$ and $t$ and $I_{s,t}$ the set of all inhibitory paths between $s$ and $t$.

Finally, we define the problem as follows: given an unweighted directed graph $G = (V,E)$, a subset of vertices $D \subset V$, representing drugs, and a subset $T \subset V$, representing target phenotypes, we are interested in finding the relative effect of a node $s$ over a node $t$ $\delta_k(s,t) = \frac{|\epsilon_{s,t}|}{|P_{s,t}|}$, for s∈D, t∈T and $1 \leq k \leq |E|$, where $\epsilon_{s,t}$ is equal to $A_{s,t}$ or $I_{s,t}$, depending on the effect we are interested in. For instance, if we want to investigate whether a drug could reverse a phenotype, we would compute the proportion of inhibitory paths over all paths of length less than or equal to $k$ between the pair of nodes.

## Algorithm

From the previous definition of relative effect, (e.g., $\delta_k(s,t) = \frac{|I_{s,t}|}{|P_{s,t}|}$ for the relative inhibition), its computation requires that activatory and inhibitory paths between nodes $s$ and $t$ are counted independently. The number of paths from $s$ to $t$ with length less than or equal to $k$ can be defined as the sequence shown in Eq 1. From the equation, it is intuitive to think of a recursive implementation to traverse the graph using a modified version of the DFS (Depth First Search) algorithm. This definition yields the foundations for an intuitive yet optimized algorithm by means of *dynamic programming* and *memoization*.

$$all_{paths}(s,t,k) = \{1 \; if \; s = t, otherwise \sum path \; s(u,t,k-1) \forall u \in neighbors(s)\} \qquad Eq \; 1$$

Dynamic programming is a method for solving a complex problem by breaking it down into simpler problems whose solutions are part of the former's solution. From Eq 1, we can easily extract that the problem of finding the number of paths from $s$ to $t$ can be broken down to finding the number of paths from all neighbors of $s$ to $t$, with maximum length of $k$-$1$. Once a solution for *all_paths(u, t, k)* is found, for any $u \in V$, it is stored and used whenever it is a subproblem to be solved again. This optimization technique is called *memoization* and is what guarantees that a node is never revisited with the same length $k$.

We have implemented two variants of drug2ways to calculate the relative effect of a pair of nodes, namely *all_paths* and *all_simple_paths* (detailed explanation and pseudocode in the S1 Text). The former considers all paths between two nodes in the graph, i.e. including *cyclic paths*, while the latter considers only *simple paths* (Table 3). This differentiation is important because *all_simple_paths* adds the restriction that cycles must be avoided and with it comes a higher complexity of the algorithm, as some nodes might be revisited. In order to evaluate the scalability of our methodology with respect to comparable methods for graph traversal, in the Subsection *Performance comparison and scalability of the algorithm*, we analyzed the performance of two variants of drug2ways (i.e., *all_paths* and *all_simple_paths*) to obtain the number of activating and inhibiting paths between pairs of nodes. We then compared the performance of drug2ways against two equivalent path-finding methods implemented in two state-of-the-art network libraries.

## Complexity

Both variants of drugs2ways (i.e. *all_paths* and *all_simple_paths*) traverse the graph visiting nodes recursively in DFS order with a maximum path length *k*. However, as previously stated, reasoning over all paths versus only simple paths are two different problems with disparate computational complexities. In the first variant of drug2ways (i.e., *all_paths*), once a node is visited, it is never revisited for a path length less than or equal to *k*, as the intermediate result stored in the cache is enough to guarantee a valid solution ([S1 Text](): Algorithm 1). In the worst case, a node is visited *lmax* times, with *lmax* being the maximum path length when the algorithm starts. Therefore, *all_paths* has a complexity of $O(lmax \times |V|)$. As for space, the cache stores two integer values (activatory and inhibitory paths are counted separately) for each pair of nodes *u*, *t* for $u \in T^C \subset V$ and $t \in T \subset V$ and for each length $k$ $1 \leq k \leq |E|$, for which a node has been visited. This translates to an upper bound in space of $\frac{|V|^2}{4}$. Thus, the algorithm has a space complexity of $O(|V|^2)$. Nevertheless, we would like to note that for biological graphs and the applications of the algorithm we devised, it is rarely the case that every target node in the graph is explored. As a consequence, the complexity is lower on the average case, as the number of target nodes is usually a small subset of *V* and the number of targets to explore is in the order of units. On the other hand, the second variant of drug2ways (i.e., *all_simple_paths*) revisits a node every time a cycle is detected ([S1 Text](): Algorithm 2). This increases the complexity to $O(|V|^{lmax})$ in the worst case. However, the average case is still several orders of magnitude faster than other standard algorithms, as discussed in the Subsection *Performance comparison and scalability of the algorithm*.

## Software and implementation

To facilitate the usage of the algorithm presented in the previous section, we implemented it in a Python package called drug2ways. The package leverages state-of-the-art Python packages such as NetworkX for network analysis [35], MPI for parallelization (https://mpi4py.readthedocs.io/), and click for exposing the command line interface (CLI) (https://click.palletsprojects.com). Drug2ways allows users to use the algorithm on a variety of standard network formats (e.g., GraphML, Node-Link, and EdgeList) and is powered by a CLI, following the standard proposed by [38]. The CLI offers all the case scenarios for proposing drug candidates that are presented in the results section.

The Python package is available at https://github.com/drug2ways/drug2ways, its latest documentation can be found at https://drug2ways.readthedocs.io and its distributions can be found on PyPI at https://pypi.org/project/drug2ways. Finally, the scripts for generating the figures in this manuscript are included in Jupyter notebooks at https://github.com/drug2ways/results.

## Case scenarios

**Networks.** To demonstrate the above-mentioned applications, we used two different multimodal networks of varying size and content ([Fig 4]()). Although each of the two networks contain unique interactions depending on the source databases they include, both the networks incorporate the following types of relations: drug-protein, protein–protein, protein–indication, and protein–phenotype. Minimally, we required each of these relationships as they simulate the binding of a drug to a target (i.e., drug-protein relation), the triggering of a cascade of events (i.e., a set of protein-protein interactions), and an effect on an indication or a phenotypic observation (i.e., protein-phenotype/indication associations), respectively. Notably, while all relationships maintained their original directionality from their source database, protein-

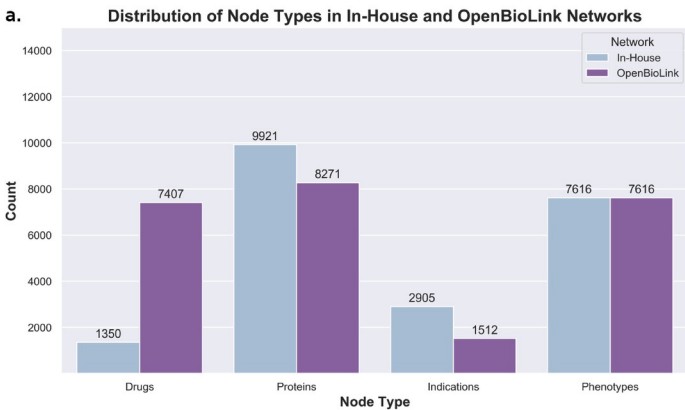

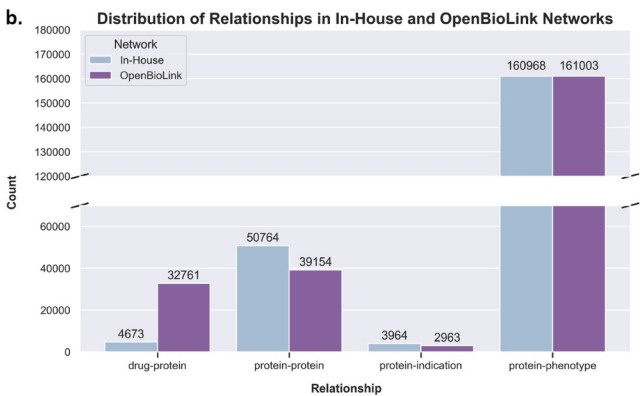

**Fig 4. Distribution of node types and relationships in the In-House and OpenBioLink networks. a)** The OpenBiolink KG contains a greater proportion of PubChem drugs relative to the In-House network which solely contains drugs from DrugBank. While the number of proteins in each of the two networks is comparable, indications are more numerous in the In-House network with respect to the OpenBioLink KG. Phenotypes for the In-House network were sourced from OpenBioLink, and as such, are equivalent in number. **b)** The total number of drug-protein interactions is greater in the OpenBioLink network than in our In-House. A greater proportion of protein-protein interactions are present in the In-House network, as are the number of protein-indication edges while the number of protein-phenotype interactions are nearly equivalent.

phenotype and protein-indication associations lacked explicit causal information and were thus inferred as activation relationships. Details about the types of each interaction are provided in S1 and S2 Tables. Below, we describe each of the two networks used.

The first network, OpenBioLink, is a large-scale KG generated from an integrative effort designed to establish a benchmark dataset for link prediction [39]. The second is an In-House network that is comprised of tens of thousands of interactions from eight databases that we have harmonized for this work including PathMe [40–43], BioGrid [44], IntAct [45], and PathwayCommons [46] for protein-protein relations, DrugBank [47] for drug-protein relations, and DisGeNet [26] for protein-indication interactions. In addition to these eight databases, protein-phenotype relationships were sourced from the OpenBioLink KG.

**Validation experiments.** In the first of three validation experiments, we ran the algorithm on two versions (all paths vs simple paths) of each of the networks over a wide range of *lmax*. We selected 2 as the minimum *lmax* as we require at least one intermediate target node between a drug and an indication. In choosing 2 as the lower bound, we incorporate the shortest possible path between a drug and an indication. However, our approach was focused more heavily on elaborate paths as a means to exploit a greater degree of complexity in biological networks. Accordingly, we set 8 as an upper bound for *lmax* such that longer paths connecting a target and a disease could also be explored. Above this range, the score, defined as the proportion of activatory/inhibitory paths (i.e., activation/inhibition ratio) tends to converge as the effect of a drug appears to cancel itself out through several, contradictory interactions (S2 Text and S1 Appendix). This event is altogether unsurprising and could be partially explained by interactions that may not be biologically plausible and through the exploration of distant nodes. Thus, users that intend to use our methodology on a different network should first study the distribution of scores as *lmax* increases, prior to determining an optimal *lmax* range. The reason is that an optimal *lmax* range can vary depending on the characteristics of a network (e.g., size, number of activation versus inhibition interactions, average number of connections, etc). Finally, we would also like to mention that a significant increase in computational time would be required for the algorithm to run for larger values of *lmax* as the number of paths with an *lmax* of 8 exceeds several millions for numerous drug-disease pairs (see Subsection *Performance comparison and scalability of the algorithm*).

**Table 4. Clinical trial information mapped to the OpenBioLink and In-House networks for drug2ways validation.** The procedure to extract the information from ClinicalTrials.gov and the corresponding lists of drugs and diseases are available at **https://github.com/drug2ways/results/tree/master/validation**.

| Network | Drug-Disease Pairs from ClinicalTrials.gov | Unique Drugs | Unique Diseases | Possible Combinations |
|---|---:|---:|---:|---:|
| OpenBioLink | 5.151 | 610 | 264 | 161.040 |
| In-House Network | 9.537 | 671 | 378 | 253.638 |

In the second experiment, we sought to validate drugs which could be effective against a given disease by incorporating clinical trial information in line with similar recent validation approaches in the literature [48,49]. As clinical trial investigations evaluate the effects of drug interventions for various indications, drug-disease pairs from ClinicalTrials.gov were used as the ground-truth list of positive labels. In total, 59.798 unique drug-disease pairs were extracted from the ClinicalTrials.gov website on 16-04-2020. Since our approach will only find paths between pairs when both the drug and disease are present in the network, only those pairs from Clinicaltrials.gov that could map to OpenBioLink and the In-House network were used as positive labels (Table 4). Thus, the original list of 59.798 unique drug-disease pairs was reduced according to the number of pairs that could be mapped to each network (i.e., 5.151 for OpenBioLink, and 9.537 for the In-House network). To conduct the validation experiments, we ran drug2ways using the drugs (source nodes) and the diseases (target nodes) present in these two filtered lists of positive labels, corresponding to a total of 161.040 possible pairs for OpenBioLink and 253.638 for the In-House network (Table 4).

Our approach exhibits the so-called early retrieval problem, or in other words, from the thousands of possible combinations of drug-disease pairs, only the top-ranked pairs contain interesting candidates for drug discovery. For such classification tasks, conventional metrics such as receiver operating characteristic (ROC) curves (i.e.. AUC-ROC and AUC-PR) become inadequate [50] This is because a classifier may accurately predict positive cases in the top-ranked pairs, but have a low predictive performance in the remaining cases that are not particularly interesting for drug discovery, leading to Area Under the Curve (AUC) values close to 0.5. For example, imagine a scenario in which 150.000 combinations of drug-disease pairs are possible in the OpenBioLink network, and of these, 5.000 are positive labels (i.e., 3%). From all possible combinations, if we consider the top 100 pairs prioritized by drug2ways and of these, 50 (i.e., 50%) are true positives, then drug2ways has captured a significantly greater number of true positives (50%) than what is expected by chance (3%). However, depending on the ranking of these pairs, it is possible to obtain a low AUC-ROC if the true positives are fairly distributed across this list of 100 pairs. Furthermore, some of these prioritized pairs may represent potential drug-disease pairs that have not been investigated before. Finally, we would like to note that only 3% of drug-disease pairs are positive labels in both networks; thus, implying a significant imbalance of class labels (Table 4). In light of these shortcomings, we have evaluated drug2ways using the AUC-ROC as a metric, yielding an AUC value of approximately 0.65 for both networks and versions of the algorithm (S3 and S4 Figs). Nonetheless, we also present a validation based on the ratio of true positives that appear in the top-ranked drug-disease pairs in order to evaluate the top-ranked set of pairs prioritized by drug2ways. Subsequently, we prioritized these pairs if they fulfilled the prioritization criteria as follows (see examples in Table 5):

1. **High inhibition.** Since we are interested in identifying drugs that inhibit a particular indication, for a pair to be prioritized, we required that at least 75% of the paths between the pair must be predicted to inhibit the indication. As we empirically selected this value, we also studied the effect of this parameter on the performance of drug2ways in the S5 and S6 Tables.

**Table 5. Illustration of the prioritization with three example pairs (i.e., A, B, and C).** For each *lmax*, the number and percentage of inhibitory paths is shown. While all three pairs show a similar pattern, pair B has less than 70% of inhibitory paths for *lmax* = 3 (i.e., Criterion 2) while for pair C, an increase in the number of paths from *lmax* = 2 to *lmax* = 3 does not occur (i.e., Criterion 3). Finally, pair A fulfills all three criteria and can thus be categorized as a prioritized pair.

| Pair | lmax | | | | | | | Prioritize |
|---|---|---|---|---|---|---|---|---|
| - | 2 | 3 | 4 | 5 | 6 | 7 | 8 | - |
| A | 1 (80%) | 4 (90%) | 20 (100%) | 50 (100%) | 100 (80%) | 400 (90%) | 1.000 (80%) | Yes |
| B | 1 (80%) | **4 (70%)** | 20 (100%) | 50 (100%) | 100 (80%) | 400 (90%) | 1.000 (80%) | No |
| C | 1 (80%) | **1 (90%)** | 20 (100%) | 50 (100%) | 100 (80%) | 400 (90%) | 1.000 (80%) | No |

2. **Consistent inhibition.** The second criteria aimed at testing the stability of the predicted effect for a given pair independent of changes to *lmax*. Accordingly, we only consider pairs where the previous criteria (i.e., more than 75% of the paths inhibit the disease) is maintained through the *lmax* range used (i.e., from 2 to 8).

3. **Increasing number of paths.** With each incremental increase in *lmax*, the number of paths must also increase such that novel paths are reported at every step of *lmax*.

As a third and final validation, we compared the two prioritized lists for each network against random lists generated by permuted versions of the original networks that were created using the XSwap algorithm [51]. By using this algorithm, we ensured that the permuted versions preserved the original structure of the original network (i.e., each node has the same number of in- and out-edges) as well as maintained the same number of activation and inhibition edges.

## Hardware

Computations for each of the tasks were performed on a symmetric multiprocessing (SMP) node with four Intel Xeon Platinum 8160 processors per node with 24 cores/48 threads each (96 cores/192 threads per node in total) and 2.1GHz base / 3.7 GHz Turbo Frequency with 1536GB/1.5TB RAM (DDR4 ECC Reg). The network was 100GBit/s Intel OmniPath, storage was 2x Intel P4600 1.6TB U.2 PCIe NVMe for local intermediate data and BeeGFS parallel file system for Home directories.

## Supporting information

**S1 Fig. Frequencies of the lengths of the shortest-paths calculated between all drug-disease pairs with lmax < = 8 in the OpenBiolink and In-House networks.**
(DOCX)

**S2 Fig. Distribution of total paths between all drug-disease pairs in the OpenBiolink and In-House networks with lmax = 8.**
(DOCX)

**S3 Fig. The AUROC curves for both networks presented in the case scenario using the all paths version of drug2ways.**
(DOCX)

**S4 Fig. The AUROC curves for both networks presented in the case scenario using the simple paths version of drug2ways.**
(DOCX)

**S1 Table. Relationships in the In-House network and their assigned polarity.**
(DOCX)

**S2 Table. Relationships in OpenBioLink and their assigned polarity.**
(DOCX)

**S3 Table. Results of the validation experiments focusing on prioritized drugs that activate an indication.**
(DOCX)

**S4 Table. Phenotypes associated with cystic fibrosis of pancreas, the indication investigated in the Subsection** *Identifying drug candidates with multiple phenotypic targets*.
(DOCX)

**S5 Table. Effect of the percentage of inhibitory paths on the number of true positives (6/7 lmax inhibit).**
(DOCX)

**S6 Table. Effect of the percentage of inhibitory paths on the number of true positives (7/7 lmax inhibit).**
(DOCX)

**S1 Appendix. "score_distributions.zip".** Distribution of the scores for each *lmax* value on both networks
(ZIP)

**S1 Text. Algorithm.**
(DOCX)

**S2 Text. Comparing distribution scores between the original and permuted networks.**
(DOCX)

## Acknowledgments

The authors would like to thank Sophia Krix for her assistance generating the networks and Colin Birkenbihl for his valuable feedback.

## Author Contributions

**Conceptualization:** Daniel Domingo-Fernández.

**Data curation:** Sarah Mubeen, Daniel Domingo-Fernández.

**Formal analysis:** Daniel Rivas-Barragan, Sarah Mubeen, Daniel Domingo-Fernández.

**Funding acquisition:** Francesc Guim Bernat, Martin Hofmann-Apitius, Daniel Domingo-Fernández.

**Investigation:** Daniel Rivas-Barragan, Sarah Mubeen, Daniel Domingo-Fernández.

**Methodology:** Daniel Rivas-Barragan, Daniel Domingo-Fernández.

**Project administration:** Francesc Guim Bernat, Martin Hofmann-Apitius, Daniel Domingo-Fernández.

**Resources:** Daniel Rivas-Barragan, Sarah Mubeen, Daniel Domingo-Fernández.

**Software:** Daniel Rivas-Barragan, Daniel Domingo-Fernández.

**Supervision:** Daniel Domingo-Fernández.

**Validation:** Daniel Rivas-Barragan, Sarah Mubeen, Daniel Domingo-Fernández.

**Visualization:** Daniel Rivas-Barragan, Sarah Mubeen, Daniel Domingo-Fernández.

**Writing – original draft:** Daniel Rivas-Barragan, Sarah Mubeen, Daniel Domingo-Fernández.

**Writing – review & editing:** Daniel Rivas-Barragan, Sarah Mubeen, Daniel Domingo-Fernández.

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
