## [Decision Letter · Decision Letter 0]

9 Sep 2020

Dear Mr Domingo-Fernández,

Thank you very much for submitting your manuscript "Drug2ways: Reasoning over causal paths in biological networks for drug discovery" for consideration at PLOS Computational Biology.

As with all papers reviewed by the journal, your manuscript was reviewed by members of the editorial board and by several independent reviewers. In light of the reviews (below this email), we would like to invite the resubmission of a significantly-revised version that takes into account the reviewers' comments.

We cannot make any decision about publication until we have seen the revised manuscript and your response to the reviewers' comments. Your revised manuscript is also likely to be sent to reviewers for further evaluation.

Sincerely,

James R. Faeder

Associate Editor

PLOS Computational Biology

Jason Papin

Editor-in-Chief

PLOS Computational Biology

Reviewer's Responses to Questions

**Comments to the Authors:**

Reviewer #1: Rivas-Barragan et al. present durg2ways, a method to calculate the proportion of pathways that possibly yield activation (or inhibition) among all possible pathways of length of at most k. The approach enumerates all paths with and without passing from cycles (refered as simple vs all paths, respectively) and combines the effect of genes on a path simply by multiplying the signs (+1 for activation, -1 for inhibition) of the pairwise relationship. The authors apply the proposed method on assessing the recovery of drug (and drug combination)-disease pairs in clinical trials.

The article is clearly written and aims to address an important problem in systems biology. That being said, in its current form it fails to offer sound evidence that support the validity of the claims in terms of the usability of the method for drug discovery. Accordingly, the text reads as a technical description of the algorithm without providing any biological insights. In particular:

1. The computational framework for reasoning causality over the paths of the network makes a very strong assumption that rarely hold in a cellular context: The regulatory / functional relationships between genes are assumed to be equally weighted, failing to account for the context-specificity (across different tissues, biological processes and individuals) of such kind of interactions. The authors could have a look at Vinayagam et al 2014 Nat Meth pmid: 24240319 for a signed PPI. It is also common practice to use gene (co-)expression as a proxy for weights between pairs of nodes (see Fakhry et al 2016 BMC Bioinf pmid: 27553489).

2. All of the case studies lack a proper (cross-) validation in regard to how well the method is able to predict drug (combination)-disease pairs. It would be important to define clearly the training data set (e.g., from clinical trials or previous work that uses drug indications such as Gottlieb et al 2011 Mol Sys Bio pmid:21654673, Guney et al 2016 Nat Comm pmid: 26831545) and use standardized metrics. I understand that due to the unbalance in the data and existence of possible repurposable drug-disease pairs (that are considered false positives in current training sets) AUROC might not be the best metric, nevertheless along with AUPRC (area under precision-recall curves) they do still provide a fair picture of the predictive performance of the method across different thresholds (e.g., proportion of activating/inhibiting paths to all paths).

3. In relation to my previous comment, though the authors compare the method to the enumeration of paths using two graph analysis libraries, given the biological problem the tool tires to address, a comparison to the tools that aim to predict the drug-disease pairs needs to be added. In fact some of the algorithms in GUILDify suite (Guney et al 2012 Plos One pmid: 23028459, Aguirre-Plans et al 2019 JCB pmid: 30851278) do simulate paths of length at most k, similar to the proposed method. Other methods exploring the use of alternative paths have also proposed (see Shahreza et al 2017 Brief in Bionf pmid: 28334136 ).

4. Similarly, if I understand correctly from Table1 (section 2.1), the performance of the method is poor, almost as good as random selection (50%). In my opinion this case study should be clarified substantially. It is also unclear what was the top-ranked number of drug-disease pairs used in this analysis.

5. Section 2.2 lacks negative controls, that is for how many of the remaining drugs (among all the drugs tested in clinical trials) possess paths to the targets.

6. Section 2.3 should also be revised to explain how exactly these combination treatments are selected and what is the significance of finding these combinations among all the possible combination treatments.

Reviewer #2: Reproducible report has been uploaded as an attachment.

Reviewer #3: Authors introduce a network-based approach for drug repurposing in diseases. Mainly, they leverage all possible paths and simple paths in a network composed of drug-protein, protein-protein and protein-indication/phenotype interactions where the source is drugs and targets are indication/phenotype. Although the subject and the problem that authors approach are very important, there are some shortcomings and major points that should be addressed. These points are listed below:

- The details of the method and the prediction approach for drug candidates are very hard to follow in the manuscript. For example, authors state in Figure 1 that “In the example, we want to investigate whether one of the three drugs depicted inhibits an indication and its two phenotypes. While all three drugs target the disease, two of the three (i.e., drug A and C) fail to produce the desired effects (i.e., inhibition of the indication of interest and its two associated phenotypes). However, drug2ways predicts that drug B could be a promising candidate as the majority of the paths between the drug and three target nodes of interest (i.e., indication and its phenotypes) would result in their inhibition, thus producing a therapeutic effect.” However, how they determine if the relation between protein and indication or phenotype is inhibitory or activating, is missing or lost in the text. In the methods part, authors describe the theoretical background, algorithm and its complexity to find paths, but the description of the validation dataset, the method to calculate “Normalized scores of the relative effects of drugs”, parameter tuning approach are missing from the main text.

- In the performance evaluation on page 6, “OpenBioLink showed good results (i.e., ~50% and ~10% recovery rate for all paths and simple paths, respectively)” is written. But, what is the definition of “good”?

- Performance comparison of the method is between the permuted networks vs original networks. An additional comparison is only based on running time of other path calculation methods. However, I can not assess from the manuscript if this drug2ways perform better that other available methods in the literature that target predicting drug candidates. A comparative performance evaluation with other available equivalent methods is necessary.

- I do not think the filtering criteria is described in subsection 4.5.2. What is the purpose of this filtering? As in the whole manuscript, this type of critical information and their discussion is very limited.

- In Figure 1, an example having a cycle and the associated simple paths are given. However, visiting a node multiple times is not so feasible in biological systems. This can be added as a constraint to the method.

In general, the current state of the manuscript is not reader-friendly. There are so many parts that actually needs to go to the results part and as explained above the method details are missing. The design and flow of the manuscript needs to be rigorously reviewed and the lacking details should be completed.

Minor points:

Figure 2b. Column names in the heatmap are not properly aligned.

Page 13 typo “…. by means of dynamic programming and memoization .”

**Have all data underlying the figures and results presented in the manuscript been provided?**

Reviewer #1: Yes

Reviewer #2: Yes

Reviewer #3: Yes

PLOS authors have the option to publish the peer review history of their article (what does this mean?). If published, this will include your full peer review and any attached files.

Reviewer #1: **Yes: **Emre Guney

Reviewer #2: **Yes: **Anand K. Rampadarath

Reviewer #3: No
---

## [Decision Letter · Decision Letter 1]

15 Oct 2020

Dear Mr Domingo-Fernández,

Thank you very much for submitting your manuscript "Drug2ways: Reasoning over causal paths in biological networks for drug discovery" for consideration at PLOS Computational Biology. As with all papers reviewed by the journal, your manuscript was reviewed by members of the editorial board and by several independent reviewers. The reviewers appreciated the attention to an important topic. Based on the reviews, we are likely to accept this manuscript for publication, providing that you modify the manuscript according to the review recommendations.

Sincerely,

James R. Faeder

Associate Editor

PLOS Computational Biology

Jason Papin

Editor-in-Chief

PLOS Computational Biology

[LINK]

Reviewer's Responses to Questions

**Comments to the Authors:**

Reviewer #1: I thank authors for addressing most of the points I had raised earlier, improving the manuscript substantially. That being said, in relation to some of the comments I had made in the previous version, the added value the method brings has to be clarified further in my opinion:

1. I understand that the information on signed / directed interactions are scarce in human, yet recent studies have increased the coverage of these interactions and have shown that they are useful in the characterization of drugs and diseases (Vinayagam et al. 2016 PNAS, doi: 10.1073/pnas.1603992113; Silverbush and Sharan 2019 Nat Commun doi: 10.1038/s41467-019-10887-6).

2. I agree with the authors that all classifier evaluation metrics have their strengths and limitations. To understand the prediction capacity of a model, one should not rely on only a single metric such as AUROC which is rather conservative when positive data is significantly less than negative data. Nevertheless, it still provides a framework to compare different parameters / classifiers and allows standardization of performance evaluation of methods in the literature (Lever et al. 2016 Nat Methods, doi: 10.1038/nmeth.3945). Like I had mentioned before, it is common practice to use ROC curve (as a whole or partially to only the bottom left part to address early retrieval problem as mentioned) or the area under Precision-Recall curve (AUPRC) in the case of data imbalance (Saito and Rehmsmeier 2015 PLOS ONE, doi:10.1371/journal.pone.0118432).

3. The authors could check GUILD, the standalone version that can be used to run programmatically for any given network and input set of nodes (Guney and Oliva 2012 PLOS ONEdoi: 10.1371/journal.pone.0043557).

4. (Minor) The formulas in the methods section do not display properly (at least using the PDF viewer I have used).

Reviewer #2: The Reproducibility report has been uploaded as an attachment.

Reviewer #3: The revised manuscript is significantly improved in terms of its flow and with the additional results and explanations. I have only two major concerns remained that needs to be addressed.

1. My first comment is about the novelty of the method. Authors describe drug2ways as “a novel methodology that leverages multimodal causal networks for predicting drug candidates”. I think that they need to explain the aspects of novelty at least in the Discussion parts in a detail, because there are multiple works using simple paths or shortest paths or all simple paths from target to source in exploring the impact of drugs at network level. In the manuscript there are some traces that authors mention about the novelty but it must be discussed in more details about the advantages of the method in a focused way. The limitations and future additions of the method are solidly described in the Discussion and I think the novelty aspect should be also discussed extensively by referencing to the current literature in the Discussion.

2. My second comment is about the tuning of the parameters. As I understood from the manuscript, the parameters of the model are the path length (k) and the percentage of inhibitory paths. Authors test multiple k values and nicely details its output in prediction, however the percentage of inhibitory paths value is not tuned anywhere in the manuscript. The value is constant as 75%. How is that value determined? If that would be tuned in an interval how the performance of the method would change? Or, if they have already done this and determined 75% accordingly, I would like to see the performance evaluation at least in the Supplementary Material.

A minor point is that mathematical symbols in the equations are missing in the Methods part of the revised manuscript.

**Have all data underlying the figures and results presented in the manuscript been provided?**

Reviewer #1: Yes

Reviewer #2: None

Reviewer #3: Yes

PLOS authors have the option to publish the peer review history of their article (what does this mean?). If published, this will include your full peer review and any attached files.

Reviewer #1: **Yes: **Emre Guney

Reviewer #2: No

Reviewer #3: No
---

## [Editor Report · Decision Letter 2]

23 Oct 2020

Dear Mr Domingo-Fernández,

We are pleased to inform you that your manuscript 'Drug2ways: Reasoning over causal paths in biological networks for drug discovery' has been provisionally accepted for publication in PLOS Computational Biology.

Best regards,

James R. Faeder

Associate Editor

PLOS Computational Biology

Jason Papin

Editor-in-Chief

PLOS Computational Biology

---

## [Editor Report · Acceptance letter]

18 Nov 2020

PCOMPBIOL-D-20-01171R2 

Drug2ways: Reasoning over causal paths in biological networks for drug discovery

Dear Dr Domingo-Fernández,

I am pleased to inform you that your manuscript has been formally accepted for publication in PLOS Computational Biology. Your manuscript is now with our production department and you will be notified of the publication date in due course.

With kind regards,

Nicola Davies
